# **Assessing Regional Climate Model Sensitivity to Vegetation Dynamics Informed by Remote Sensing**

Thomas Dethinne<sup>1</sup>, Nicolas Ghilain<sup>2,3,4</sup>, Christoph Kittel<sup>2,5</sup>, Benjamin Lecart<sup>1</sup>, Xavier Fettweis<sup>2</sup>, and François Jonard<sup>1</sup>

**Correspondence:** Thomas Dethinne (tdethinne@uliege.be)

**Abstract.** Climate change significantly impacts vegetation ecosystems, and their modification may create feedback loops exacerbating regional effects of global warming. Accurately simulating vegetation dynamics and their interactions with the atmosphere is crucial for understanding and mitigating these impacts. Regional earth system models offer the possibility to study the retroaction between the atmosphere and the vegetation at regional to continental scale by incorporating vegetation dynamics in climate models. In this study, we quantify the sensitivity of the Modèle Atmosphérique Régional (MAR) to vegetation representation at daily to annual scales over a temperate region of Europe, by means of both synthetic experiments and realistic studies.

Our sensitivity study on the Leaf Area Index (LAI) dynamics reveals non-linear responses on meteorological variables, with asymmetric effects relative to the direction of the change. For example, a 92 % reduction in LAI led to an 83.4 % decrease in evapotranspiration and an 88.9 % drop in evaporation. Conversely, a 178.4 % increase in LAI resulted in smaller, yet significant, increases of 29.8 %, 27.4 % respectively. At the seasonal-time scale, evapotranspiration and albedo have the strongest shifts in summer and winter, while relative humidity and rainfall responded more prominently in spring.

Furthermore, we assessed the model performance in simulating daily evapotranspiration and daily maximum temperature during extreme events by comparing simulations incorporating 8-day MODIS LAI data with those based on climatological LAI. Although the improvements were more subtle than those resulting from a change in LAI source, the 8-day observation-based LAI enhanced the model's capacity to capture shorter events compared to the static LAI. This refinement also helped understanding how various vegetation types respond to extreme events.

The findings highlight the need to integrate dynamic vegetation into regional climate models to enhance their representation of biosphere-atmosphere interactions and provide more accurate tools to assess the impacts of climate change on natural ecosystems.

<sup>&</sup>lt;sup>1</sup>Earth Observation and Ecosystem Modelling Laboratory, SPHERES Research Unit, University of Liège, Liège, Belgium

<sup>&</sup>lt;sup>2</sup>Laboratory of Climatology, SPHERES Research Unit, Department of Geography, University of Liège, Liège, Belgium

<sup>&</sup>lt;sup>3</sup>Royal Meteorological Institute, Uccle, Belgium

<sup>&</sup>lt;sup>4</sup>Unit for Modelling of climate and Biogeochemical Cycles, SPHERES Research Unit, University of Liège, Liège, Belgium

<sup>&</sup>lt;sup>5</sup>Physical geography research group, Department geography, Vrije Universiteit Brussel, Brussels, Belgium

#### 1 Introduction

Vegetation plays a crucial role in regulating Earth's climate by influencing the carbon cycle, water cycle, and surface energy balance (Liu et al., 2006). Forests, in particular, act as significant carbon sinks, sequestering carbon dioxide from the atmosphere and mitigating the greenhouse effect (Heinrich et al., 2023). However, as regional effects of climate change accelerates, these ecosystems are becoming increasingly vulnerable to disturbances (Gonzalez et al., 2010; Knutzen et al., 2025) and are likely to experience changes in their spatial distribution in the future (Roebroek et al., 2025). Extreme events such as droughts, wildfires, and floods can severely affect ecosystem health (Guion et al., 2022).. During droughts, warmer temperatures and reduced water availability increase vegetation stress (McDowell et al., 2008), both directly through physiological impacts (e.g., Jonard et al., 2011) and indirectly through pest outbreaks such as bark beetle infestations (e.g., Kolb et al., 2019). These stresses can lead to defoliation (Carnicer et al., 2011), elevated mortality rates (Allen et al., 2015), and large-scale die-off events (ying Huang et al., 2019). n response, plants may close their stomata to limit water loss and draw water from deeper soil layers, altering transpiration and soil evaporation rates (He et al., 2022). Such conditions also change surface albedo and modify land-atmosphere exchanges of energy and latent heat at regional scales (Dirmeyer et al., 2006). The resulting ecosystem degradation can amplify global warming, creating positive feedback loops that accelerates decline (Strengers et al., 2010). Consequently, monitoring natural ecosystems and simulating their future conditions are essential for supporting sustainable management and contributing to effective climate change mitigation (Verma et al., 2025). Research on surface processes advances our understanding of vegetation-climate feedbacks and informs policymakers and land managers about the potential consequences of land-use changes.

40 Climate evolution has been studied using climate models to simulate past, present, and future states over the Earth (Vogel et al., 2017; on Climate Change, IPCC; Church et al., 2013; Balcha et al., 2023). These models range from stand-alone systems that address individual Earth system components (e.g., atmosphere, hydrosphere, biosphere) to more complex Global Climate Models (GCMs) that interactively couple multiple components, also known as Earth System Models (ESMs). Prominent ESMs are widely used for global climate assessments (on Climate Change, IPCC; Gutjahr et al., 2019; Boucher et al., 2020; EC-Earth Consortium (EC-Earth), 2020), particularly in standardized simulations like the Coupled Model Intercomparison Project phase 6 (CMIP6; Eyring et al., 2016). ESMs are used in regional studies (Cabos et al., 2020; Sein et al., 2020; Soto-Navarro et al., 2020). However, because climate change often manifests more strongly at local than global scales (Vogel et al., 2017; Stuecker et al., 2018), Regional Climate Models (RCMs) and ensembles such as EURO-CORDEX (Kjellström et al., 2018) are widely used to simulate climate dynamics at high spatial resolution in regional climate assessments (e.g., Giorgi, 2019; Termonia et al., 2018; Gallée and Schayes, 1994; Yang et al., 2020; Haberlie et al., 2022; Giorgi et al., 2012; Powers et al., 2017; Zittis et al., 2022). Despite often being primarily developed to simulate only one component of the environmental system, most RCMs include schemes for the others. These schemes can either involve model coupling, similar to what is done in ESMs (Zhang et al., 2014; Mikolajewicz et al., 2005; Anwar and Diallo, 2021) creating Regional Earth System Models, or rely on static datasets that serve as forcing for specific modules of the model (Heck et al., 2001; Tedesco et al., 2023). This enables

RCMs to simulate interactions between different components of the Earth system using input forcing data while modelling the interactions with the primary compartment of the model.

Among these components, the land surface plays a crucial role, acting as an interactive boundary that governs biosphere-atmosphere interactions. These interactions can either occur between the soil and the atmosphere directly or between the vegetation and the atmosphere. Biosphere changes can alter surface albedo, evapotranspiration (ET) rates, and energy fluxes. All of which are key factors in the climate system (Strengers et al., 2010), and have a direct impact on the atmosphere (Insua-Costa et al., 2022). Understanding feedbacks between climate and vegetation is critical for predicting future climate scenarios and developing effective mitigation and adaptation strategies (Heck et al., 2001; Beniston et al., 2007). While biosphere-atmosphere interactions have been studied in RCMs, their treatment varies (Giorgi and Gao, 2018). Some addressed the interaction using land surface changes (Giorgi and Gutowski, 2015), others by improving existing land surface modules (Oleson et al., 2008), or model coupling (Anwar and Diallo, 2021; Gröger et al., 2021; Heck et al., 2001; Bright et al., 2025), ultimately highlighting the importance of considering vegetation dynamics in regional climate modelling to better predict and mitigate the impacts of climate change (Giorgi and Gao, 2018; Gröger et al., 2021).

Remote sensing has become a key tool in climate modelling due to its ability to provide long, consistent time series observation of ecosystem state at large scale (Latifi and Galos, 2010; Jonard et al., 2022). With short revisit times and high spatial resolution, satellite datasets offer valuable information on essential climate variable such as vegetation cover, albedo, soil moisture, and land surface temperature (Liang, 2001; Entekhabi et al., 2010; Myneni et al., 2002; Wan and Dozier, 1996; Jonard et al., 2022). They have been widely used in global and regional climate models, including in reanalyses like ERA-5 (Hersbach et al., 2020), for model evaluation (e.g., Noël et al., 2025), and as direct inputs to drive model simulations. In addition, remote sensing base dataset can also be especially useful for assessing model performance during extreme events (Fluhrer et al., 2025).

Numerous studies have investigated the sensitivity of land surface models (either standalone or coupled to GCMs) to vegetation properties, due to their central role in regulating surface fluxes such as evapotranspiration, albedo, and roughness length (Schwinger et al., 2010; Chase et al., 1996; Chang et al., 2018). These sensitivities are often explored through prescribed changes in vegetation type or structure. However, despite community efforts, such as the Land Use and Climate Across Scales Flagship Pilot Study (LUCAS FPS), which includes studies like Davin et al. (2020) testing the sensitivity of multiple RCMs to two different vegetation scenarios, the quantification of RCM sensitivity to vegetation remains relatively rare.

In this study, we explore the sensitivity of the RCM MAR to various vegetation scenarios. While MAR is primarily an atmospheric model it has already been coupled to ice sheet (Delhasse et al., 2024), ocean (Huot et al., 2022; Jourdain et al., 2011) models, or use to force dynamic vegetation model (DVM). Its surface scheme, the Soil Ice Snow Vegetation Atmospheric Transfer module (SISVAT; Ridder and Gallée, 1998), allows for the modelling of energy and water budget using some external data of vegetation. Here, we integrate into MAR high-resolution, remote sensing-based vegetation observations that

better capture vegetation dynamics than the static datasets currently used.

By altering the input vegetation data, we aim to understand: (1) the representation of biosphere-atmosphere interactions in the model in view of future coupling with a DVM, (2) how these changes impact the model simulations of indicators used in climate services. Using remote sensing-based dataset, we explore various aspects of these impacts. Modifying the input dataset enables us to assess the model's sensitivity to its vegetation inputs but also how the assimilation of input data into the model influences the results. This work also investigates the impact of using a dynamical LAI, which is particularly valuable during extreme events.

The study region, the model and its vegetation inputs, and the different experiments of sensitivity are presented in Sect. 2.

The results of he assessment of the model and the sensitivities are discussed in Sect. 3. Finally, a discussions about the model and its sensitivity and perspectives of coupling the MAR model with a DVM included in Sect. 4, and a general conclusion is presented in Sect. 5.

### 2 Model and experiments

110

120

## 2.1 Regional Climate Model

The RCM used in this study is MAR (MAR model, 2025), an atmospheric model based on a hydrostatic approximation of the primitive equations of the atmosphere as described by Gallée and Schayes (1994). Its radiative transfer scheme is derived from Morcrette (2002). Originally developed to model polar climates, MAR has primarily been used for research on Greenland (Maure et al., 2023; Glaude et al., 2024) and Antarctic ice sheets (Dethinne et al., 2023; Kittel et al., 2022), and is also been applied to climate studies in Europe (Doutreloup et al., 2022; Grailet et al., 2025; Brajkovic et al., 2025).

Among the different modules of MAR, SISVAT simulates the transfer of mass and energy between the atmosphere and the soil. Within the model, three sub-pixels (called sectors) are defined to represent the two predominant vegetation types as well as grass in each pixel. MAR is configured to resolve the first 7 meters of soil, divided into 7 layers (the layers have a decreasing vertical resolution from top to bottom, with the near-surface layers of a few millimetres and deepest layers a few meters thick). Within SISVAT, the plant water flow is dictated by the water potential between the soil and the plant leaves, and the turbulent transfer is computed by considering canopy-air and ground-air exchanges separately, without an explicit parametrization of environmental stress (de Ridder and Schayes, 1997). A yearly cycle for the vegetation is derived from the values that are provided to the model as an input (see section 2.3). Minimal stomatal resistance and root density are derived from values proposed in Taylor et al. (1997) and Jackson et al. (1996) respectively. Other vegetation-dependent variables like the roughness length for momentum or the stomatal resistance are based on equations described in de Ridder and Schayes (1997). The non-dynamic vegetation cover classification and its fraction are based on the International Geosphere Biosphere Programme (IGBP) surface land classification (Loveland and Belward, 1997). It has been resampled into 13 categories: barren soil, low-medium-high

crops, low-medium-high grass, low-medium-high broadleaf, and low-medium-high needleleaf. The meteorological inputs of SISVAT consist in wind speed, temperature and humidity at the level the closest to the soil, as well as downward shortwave and longwave radiations, and precipitation.

For this study, MAR was run using the same setup than Grailet et al. (2025) over a longer time period, i.e. using a 5 km pixel size over Belgium, with 24 pressure layers from surface to low stratosphere, with a 25-second time step and hourly outputs from September 2010 to December 2024. MAR was forced every 6 hours by the ERA5-reanalysis data (Hersbach et al., 2020) at its lateral boundaries, over the ocean and in altitude (in the stratosphere) where nudging is applied over 3 layers. To minimize the influence of the model initial conditions, the first four years of each simulation and uncompleted years are excluded from the analysis, resulting in a study period of 2015-2024.

The modelled area extends slightly beyond Belgium, encompassing the southern Netherlands, northern France, western Germany, and the North Sea to the north, south, east, and west of the domain, respectively as in Wyard et al. (2017). A relaxation zone of 5 pixels is removed from each lateral boundary of the domain, resulting in the domain presented in Fig. 1. In the end, only pixels intersecting the Belgian territory are used in this study. In urban areas, no vegetation is defined except hypothetical grass. Therefore, pixels classified as urban by MAR were excluded from this analysis.

## 140 2.2 Study region

The study area cover a small region including Belgium, a country in Western Europe of approximately 3,068,000 hectares. According to the National Institute of Statistics, in 2023, the land cover of Belgium was composed of 44 % agricultural land (crops and pastures), 20 % forests, and 22 % built-up areas. However, this distribution varies across the country's two main administrative regions (Statbel, 2025):

In the North, Flanders, which accounts for 45 % of Belgium's territory, is more densely urbanized with only 8 % forest cover and nearly 30 % built-up land. In contrast, Wallonia, in the South, contains the majority of Belgium's forests, which cover around 30 % of its territory, mainly concentrated in the south, in the Ardennes region (Statbel, 2025). The land cover used in this study (Fig. 1) do not match these numbers as MAR uses only the main two land cover within pixels of 25 km<sup>2</sup>, with a slight overestimation of agricultural land in spite of built-up area. For illustration reason, the land cover of Fig. 1 has been filtered using a 3x3 mode window.

Belgium's current climate is classified as Cfb in the Köppen-Geiger climate classification (Beck et al., 2018), which corresponds to a temperate oceanic climate with mild summers and relatively wet, cool winters. The mean annual temperature reaches around 10 °C and the main factor for the temperature spatial variability is caused by the change in altitude. Also, the proximity to the sea has a role in the change in temperature in the northwest of Belgium (Erpicum et al., 2018). With about 700 mm to 1300 mm of rainfall on average per year depending on the location, precipitation is fairly evenly distributed throughout


**Figure 1.** Extension of the studied domain. The names of regions of interests and these used during the study are included on the map. The background map shows the dominant land cover considered by the MAR model. For clarity, the land cover has been filtered using a 3x3 mode window.

the year. Though, seasonal variations exist, with a monthly rainfall minimum observed during the winter–spring transition (Erpicum et al., 2018).

Under this climate, Belgium's characteristic forest consists mainly of deciduous broadleaved and coniferous forests. The dominant species in Wallonia include oak, beech, and spruce, the latter being widely planted in the Ardennes for forestry (Latte et al., 2020). Low vegetation primarily consists of agricultural land (Statbel, 2025).

These differences in land cover throughout the country, the variety in vegetation type, and the relatively homogeneous topography of the region make Belgium a relatively interesting region to study the sensitivity of MAR to its input in vegetation.

During the study period, Belgium underwent multiple events of drought and pluvial periods. These periods can be identified through the Standardized Precipitation Evapotranspiration Index (SPEI; Vicente-Serrano et al., 2010), a multiscalar index using both precipitation and evapotranspiration for determining drought durations and magnitudes. Three periods of relative drought and three periods of relatively moist conditions were selected to study the behaviour of the model during stronger events. The events were selected using the average value of the SPEI index at 90 days over Belgium during the period. The drought events (SPEI < -1.5) occurred in summer 2018, 2019, and 2022. The pluvial periods (SPEI > 1) were observed during summer 2021, 2023, and 2024 (Fig. 2).

**Figure 2.** Evolution of the average 90-day SPEI3 over Belgium for the study period. Strong events of drought are easily identified in summer 2018-2019-2022 while 2024 stands out as a relatively wetter year.

#### 2.3 LAI data




As explained in Section 2.1, the vegetation cycle in MAR is externally driven. In this case, MAR use as primary information the Leaf Area Index (LAI), that represents the one-sided green leaf area of a canopy per unit ground area, influencing processes like photosynthesis, and energy exchange at the vegetation-atmosphere interface (Chen and Black, 1992).

In its standard configuration, MAR uses a climatological mean LAI, derived from the MERRA2 reanalysis (Global Modeling and Assimilation Office (GMAO), 2015; Gelaro et al., 2017) for the 1961–1990 period at 50 km spatial resolution (Wyard et al., 2021). We refer to this LAI dataset as "MERRA2 LAI" in the text hereafter. The climatology is computed as monthly LAI values, which are later linearly interpolated between to a 6-hour time step to create a continuous time series to force the vegetation model of MAR. MERRA2 LAI is integrated into MAR by linearly interpolating the values to the MAR grid using a weight proportional to the distance between the pixel centres. Then for the first two sub-pixels, a correction coefficient is applied to the LAI values, depending on the vegetation considered by MAR for that sub-pixel (Fig. 3a). The correction coefficient value is determined in order to maintain consistency in the results with the previous version of the model not using MERRA2 LAI.

For this study, we replaced MERRA2 LAI by the LAI from the Moderate-Resolution Imaging Spectroradiometer (MODIS) (Myneni et al., 2021), that is a higher-resolution dataset, both spatially, with a 500 m resolution and temporally, with an image every 8 days. This already long series of vegetation indices is convenient because it fulfils the requirements of our study: the possibility to explore spatial and temporal sensitivities over climate compatible horizons. To minimize boundary effects in the








model, the MODIS data was extracted over a region extending beyond the MAR domain. From the 8-day MODIS, we also computed a climatology using all the images available over the period 2012–2022 Only pixels with quality flags indicating confidence scores of 0 (best results) and 1 (good results) were used from the images. However, artefacts of relatively high LAI appear at the end of December as already observed in Satellite Application Facility on Land Surface Analysis LAI time series (LSA SAF; Ghilain et al., 2012b). To further refine the climatology, we applied a moving average with a 32-day window (2 images before and two images after) to smooth the LAI time series for each pixel. This climatology and the 8-day product were linearly interpolated at 6-hour intervals to match MAR forcing time step requirements. This MODIS climatology is hereafter referred to as "MODIS LAI". Since MODIS does not provide LAI values for urban pixels, we decided to set LAI to zero in these areas, given that MAR does not compute vegetation-related processes in cities.

MODIS LAI values are resampled to match the MAR 5 km grid when input into the model. The resampling for the MODIS climatology is similar to the one used for MERRA2 but without any correction coefficient as we don't need to keep the consistency in the vegetation indices with the MERRA2 version(Fig. 3b). The experiments use the same LAI values in the first two sub-pixels, excepted for one of the simulations, where the values from the MODIS climatology is adjusted given vegetation considered in the sub-pixel of MAR. The scaling operation is performed during the resampling from the 500 m MODIS pixel size to the 5 km MAR resolution by creating a mask of LAI values derived from the vegetation cover given in the IGBP surface land classification and by only using the MODIS pixel having the same vegetation than the one given by the MAR sub-pixel (Fig. 3c).

# 2.4 Model experiments

In total, we conducted 10 different simulations to assess the sensitivity of MAR to variations in its vegetation input (see Table 1). The first simulation, referred to as  $MAR_{ref}$ , was performed using the standard MAR configuration with MERRA2 LAI described above. This simulation serves as the reference scenario and to assess the model results. The nine other simulations use MODIS LAI data. One of these simulations,  $MAR_{MODIS}$ , employed the 8-day MODIS LAI while the other utilized the climatology described above.

First, the  $MAR_{MODIS_{clim}}$  simulation used the MODIS climatology process as explained in the previous section. By using MODIS LAI instead of MERRA2 LAI, we test the impact of significantly enhancing the spatial resolution of the vegetation input (from 50 km to 500 m) and improving the temporal resolution (from monthly to 8-day intervals). Considering that LAI is the only input MAR uses to simulate vegetation, these changes are expected to have a substantial impact on the model outputs.

Six simulations  $(MAR_{+2\mu})$  to  $MAR_{-2\mu}$  were designed to explore the impact of perturbations in MODIS LAI climatology. These perturbations involved adding either positive or negative Gaussian noise to the LAI values. Gaussian noise was computed and applied daily across the entire MODIS image extending beyond the MAR domain. The standard deviation of

Figure 3. Integration of the LAI dataset into MAR. (a) Integration of MERRA2 dataset. The global dataset is sliced to the domain and then interpolated to the finer MAR grid. Then the value is provided to the first two sectors of MAR by applying a correction coefficient depending on the vegetation type in the given pixel and sector of MAR. (b) Integration of MODIS dataset. The dataset is sliced to the domain and a Gaussian noise is applied to the dataset. After, the values are interpolated to the coarser MAR grid. Then, the values of LAI are provided for the first two sectors of MAR. (c) Integration of MODIS dataset per sector of MAR. The dataset is sliced to the domain and a mask of the different vegetation types is created. Then, the values are interpolated to the coarser MAR grid by only using the mask of the vegetation type in the given pixel and sector of MAR.



**Table 1.** Summary of the names of the simulation, the LAI datasets used in the simulation and the type of noise noise applied to the LAI dataset for all the simulations performed.

| Simulation name      | LAI dataset                  | Noise                          |  |  |  |
|----------------------|------------------------------|--------------------------------|--|--|--|
| $MAR_{ref}$          | MERRA2 climatology           | -                              |  |  |  |
| $MAR_{MODIS}$        | MODIS 8-day                  | -                              |  |  |  |
| $MAR_{MODIS_{clim}}$ | MODIS climatology            | -                              |  |  |  |
| $MAR_{sector}$       | MODIS climatology per sector | -                              |  |  |  |
| $MAR_{\pm 2\mu}$     | MODIS climatology            | $\pm$ Gaussian $(2\mu,\sigma)$ |  |  |  |
| $MAR_{\pm\mu}$       | "                            | $\pm$ Gaussian $(\mu, \sigma)$ |  |  |  |
| $MAR_{\pm 0}$        | "                            | $\pm Gaussian (0, \sigma)$     |  |  |  |

the noise corresponded to the mean standard deviation of LAI for that specific day, amplifying differences when LAI spatial variability is high, while avoiding excessive noise when variability is low. The mean value of the noise was set to one of five values: zero, the average LAI value for that day, twice the average LAI value for that day, minus the average LAI value for that day, or minus twice the average LAI value for that day. These values are designed to maximize the potential to boost or reduce LAI when vegetation is at its peak, i.e. when LAI values are high, and to have a smaller influence when LAI values are already low. The noise with a mean value of 0 is computed twice, once to be added to the climatology and once to be subtracted to it. In such way, this variation in vegetation density across simulations is expected to have a significant impact on the modelled climate variables. The simulation  $MAR_{-2\mu}$  was stopped in 2019 as the simulation was slowly drifting away from observations because of instability in the hydrological budget (see section 3.3).

Then,  $MAR_{sector}$  uses the MODIS LAI dataset decomposed for each sub-pixel of MAR (Fig. 3c). This experiment allows us to study how the integration of the LAI dataset can potentially impact the results of the model by comparing it to  $MAR_{MODIS_{clim}}$ . By incorporating sub-pixel variations,  $MAR_{sector}$  provides a more detailed representation of vegetation heterogeneity, which may influence key processes in MAR, especially ET.

Finally, the  $MAR_{MODIS}$  simulation is driven by 8-day MODIS LAI data, allowing the model to capture short-term vegetation dynamics. This observation-based LAI enables  $MAR_{MODIS}$  to more accurately reflect vegetation conditions during shorter meteorological events like extreme events, in comparison to the climatological LAI used in simulations such as  $MAR_{MODIS_{clim}}$ , as  $MAR_{MODIS}$  should be better suited to reflect real-time vegetation responses to environmental stressors and can be particularly relevant to study feedback mechanisms.

#### 3 Results





#### 3.1 Model assessment

Given the inherent uncertainties in the integrated physics of RCMs, it is essential to evaluate their outputs against in situ measurements to accurately assess how well the model represents real-world conditions and reproduces observed weather patterns (Brajkovic et al., 2025; Grailet et al., 2025). In this study, our main focus is on evaluating the sensitivity of the MAR model to variations in vegetation input. It is important to recognize that any deviations between model outputs and observations in such studies could be more reflective of the noise applied to the input data rather than the model core physics. In this manner, only  $MAR_{ref}$  and  $MAR_{MODIS_{clim}}$  are evaluated.

To evaluate the simulations, we compared the daily averaged model outputs against observational-modelled data. Specifically, we used data of several key meteorological variables: sea-level pressure, surface air temperature, wind speed, relative humidity, rainfall, collected from weather stations and gridded over Belgium using a statistical post-processing (Journée et al., 2023), and actual evapotranspiration estimates from LSA SAF (Ghilain et al., 2011, 2012a) to perform a statistical comparison between the model and observed data. The statistical metrics used for this evaluation include the mean bias (MB), root-mean-square error (RMSE), and Pearson correlation coefficient (r) (see Table 2). A negative MB implies an underestimation by the model. These metrics were calculated for the period from 2015 to 2020.

Overall, both simulations exhibit strong performance in capturing surface air temperature and sea-level pressure, as indicated by the high correlation coefficients (r > 0.9) and relatively low RMSE values. In addition, the model tends to better represent winter than summer, as already been documented for the precipitation (Doutreloup et al., 2019).

Rainfall is the only variable showing a correlation coefficient lower than 0.8 annually despite relatively low mean bias and RMSE. The major problem in the rainfall simulation concerns the strong rain events Simulating their exact spatial and temporal extent can be challenging and may lead to strong reduction of correlation values. de Vyver et al. (2021) and Brajkovic et al. (2025) already highlighted that MAR tends to underestimates the extreme precipitation events. However, the daily mean bias being lower than the daily variation of the observation still suggests a good representation of rainfalls.

For air temperature,  $MAR_{MODIS_{clim}}$  shows a slight improvement over  $MAR_{ref}$ , with a reduction in RMSE. This suggests that the MODIS LAI input potentially enhances the model's ability to simulate the surface energy balance more accurately during warmer months by having a better representation of the vegetation. The higher RMSE and mean bias observed in summer and winter than on an annual basis indicate that the model underestimates extreme air temperatures, possibly due to limitations in the representation of soil moisture and vegetation feedback.

**Table 2.** Mean Bias (MB), Root Mean Square Error (RMSE), and Pearson-correlation coefficient (r) between two MAR simulations  $(MAR_{ref}, \text{ and } MAR_{MODIS_{clim}})$  and daily observations over Belgium. A negative value implies a lower MAR estimate compared to the observation. Statistics are given for the sea-level pressure, surface air temperature, wind speed, relative humidity, rainfall, and actual evapotranspiration, for the summer (JJA) and winter (DJF) seasons, and annually for the 2015-2020 period. MB marked with an asterisk (\*) indicates that the mean daily MB is smaller than the daily standard deviation of the observations. A dash (-) in the MB column signifies that there is no significant difference (p-value > 0.05) between the model mean daily values and the observations. Refer to Table 1 for the simulation names.

|                      |                                | Annual |      | Winter      |        |      | Summer      |        |      |             |
|----------------------|--------------------------------|--------|------|-------------|--------|------|-------------|--------|------|-------------|
|                      |                                | MB     | RMSE | Correlation | MB     | RMSE | Correlation | MB     | RMSE | Correlation |
| $MAR_{ref}$          | Sea-Level Pressure (hPa)       | -      | 1.64 | 0.99        | -      | 2.12 | 0.99        | -      | 1.08 | 0.98        |
|                      | Temperature (°C)               | -      | 1.18 | 0.99        | -      | 1.1  | 0.96        | -      | 1.35 | 0.94        |
|                      | Wind speed (ms <sup>-1</sup> ) | -0.52* | 0.6  | 0.97        | -0.56* | 0.63 | 0.98        | -0.44* | 0.53 | 0.95        |
|                      | Relative humidity (%)          | 5.89*  | 7.86 | 0.92        | 8.32   | 9.37 | 0.89        | -      | 5.57 | 0.92        |
|                      | Rainfall (mm)                  | -1.12* | 2.85 | 0.78        | -1.61* | 3.14 | 0.78        | -      | 2.96 | 0.75        |
|                      | Evapotranspiration (mm)        | -1.1*  | 1.41 | 0.91        | -0.29  | 0.36 | 0.83        | -2.01  | 2.15 | 0.8         |
| $MAR_{MODIS_{clim}}$ | Sea-Level Pressure (hPa)       | -      | 1.64 | 0.99        | -      | 2.13 | 0.99        | -      | 1.09 | 0.98        |
|                      | Temperature (°C)               | -      | 1.08 | 0.99        | _      | 1.09 | 0.96        | -      | 1.14 | 0.94        |
|                      | Wind speed (ms <sup>-1</sup> ) | -0.54* | 0.61 | 0.97        | -0.56* | 0.62 | 0.98        | -0.49* | 0.57 | 0.95        |
|                      | Relative humidity (%)          | 6.94*  | 8.24 | 0.93        | 8.26   | 9.29 | 0.89        | 5.04*  | 6.57 | 0.92        |
|                      | Rainfall (mm)                  | -1.12* | 2.85 | 0.78        | -1.61* | 3.14 | 0.78        | -      | 2.96 | 0.75        |
|                      | Evapotranspiration (mm)        | -1.01* | 1.27 | 0.92        | -0.29  | 0.37 | 0.83        | -1.79  | 1.92 | 0.81        |

Evapotranspiration shows a consistent negative mean bias for both  $MAR_{ref}$  and  $MAR_{MODIS_{clim}}$  simulations. The lower annual RMSE in  $MAR_{MODIS_{clim}}$  suggests an improvement in the model representation of vegetation dynamics when using higher-resolution LAI data. However, the persistent underestimation could indicate a need for further refinement in the biosphere-atmosphere transfer processes within MAR. The non-significant change may be related to the use of a climatology instead of using observed LAI, and will be assessed in section 3.5.

While there is still room for improvement, the model demonstrates a strong ability to accurately represent the climate of Belgium during the period 2015-2020. Overall, the two experiments show similar results when compared over a long period with a slight improvement in the results for  $MAR_{MODIS_{clim}}$ .

## 3.2 Sensitivity to spatial resolution of the LAI input


As a first step, the sensitivity of the model is assessed by comparing the results of  $MAR_{ref}$  with those of the other simulations using a LAI climatology. Comparing  $MAR_{ref}$  with  $MAR_{MODIS_{clim}}$  gives an indication on how the model reacts to a change




**Figure 4.** (a) Mean Leaf Area Index (LAI) over Belgium in 2018 for the different simulations. The names of the LAI curves refer to the name of the simulation in which they are considered (refer to Table 1 for the description of the simulations). (b) Average daily difference in LAI in percent between the MERRA2 LAI climatology and the MODIS LAI climatology used in this study. Forest regions are indicated with black-dashed outlines

in the input dataset spatial resolution. Derived from a higher spatial resolution dataset (500 m vs 50 km), the MODIS climatology captures vegetation differences more accurately than the MERRA2 climatology. In this case, the MODIS LAI climatology generally exhibits higher LAI values compared to the MERRA2 LAI, suggesting that the current MAR configuration may be underestimating vegetation cover over Belgium (Fig 4(a)).

As shown in Fig. 4(b), LAI is doubled in certain places just north of the Ardennes while there is a decrease in forested land. Northern Belgium experiences the same evolution with an increase in LAI for agricultural land and a decrease in forested regions, such as the Campine and areas south of Brussels of approximately 75 % and 25 % respectively.

Given that the vegetation input has been modified, it is anticipated that variables directly related to vegetation, such as ET, will reflect these changes.  $MAR_{MODIS_{clim}}$  exhibits higher ET over the study period compared to  $MAR_{ref}$ , with an increase of from 120 mm of ET to 156 mm per day on average in summer (Table 3). This increase is explained by the higher LAI in  $MAR_{MODIS_{clim}}$  relative to  $MAR_{ref}$  which lead to higher ET from leaves. These changes are most pronounced during summer and spring when vegetation is most active then in winter and autumn where no significant changes (p-values > 0.05) are observed (Fig. 5(a)).

Impact on surface air temperature also varies between seasons (Fig. 5(b)) and remains quite small. During the summer,  $MAR_{MODIS_{clim}}$  generally show air temperatures that are slightly more than half a degree Celsius cooler as expected with a

**Table 3.** Seasonal daily average values of surface air temperature, soil moisture content, and albedo, along with cumulative mean daily rainfall and evapotranspiration, for summer (JJA) and winter (DJF) during 2015–2024, as simulated by  $MAR_{ref}$  (using the MERRA2 LAI climatology) and  $MAR_{MODIS_{clim}}$  (using the MODIS LAI climatology).

|                        | Temperature (°C) |      | Soil moisture (%) |      | Albedo |      | Rainfall (mm) |     | ET (mm) |     |
|------------------------|------------------|------|-------------------|------|--------|------|---------------|-----|---------|-----|
|                        | DJF              | JJA  | DJF               | JJA  | DJF    | JJA  | DJF           | JJA | DJF     | JJA |
| $\overline{MAR_{ref}}$ | 3.8              | 18.7 | 67.8              | 65   | 26.2   | 22.7 | 194           | 230 | 6       | 120 |
| $MAR_{MODIS_{clim}}$   | 3.8              | 18.4 | 65.7              | 62.8 | 26.3   | 23.8 | 194           | 232 | 6       | 156 |

denser vegetation. On the opposite, during winter, these experiments tend to be warmer by about  $0.2 \,^{\circ}\text{C}$  (p-values > 0.05).

However, surprisingly, the increase in ET does not translate to a change in rainfall intensity (Table 3) but in the spatial patterns with a dependence on the season. The pattern of rainfall differences is neither directly correlated to the patterns observed in LAI changes nor in the vegetation cover, implying that rainfall patterns are indirectly affected (Guillod et al., 2014; Hsu et al., 2017; Santanello et al., 2018).

In terms of soil water saturation,  $MAR_{ref}$  shows a statistically significant steady decline throughout the simulated period of 0.5 % per year that can be attributed to the inability of the model to correctly represent the soil hydrology cycle and tend to dry over the years. The highest average soil moisture saturation in Belgium was 0.72 % during spring 2015, which dropped to a minimum of 0.56 % in fall 2019. This decline also occurs in  $MAR_{MODIS_{clim}}$  and is faster (0.8 % per year). Although  $MAR_{MODIS_{clim}}$  has lower soil water saturation than  $MAR_{ref}$  for the whole studied period (Table 3), a seasonal cycle emerges in the soil moisture differences, with smaller variations in spring and more pronounced differences by late summer. This fluctuation in soil moisture is primarily driven by root water uptake, greater in simulations with more dense vegetation and reduced in those with sparser vegetation and is directly linked with the increase/decrease in ET.

Finally, the comparison is concluded by examining the components of the energy budget, including longwave upward (LWU), shortwave upward (SWU), and sensible heat (SHF) fluxes. For sensible heat, a positive value indicates a flux directed towards the Earth's surface. To simplify the analysis, we omit latent heat flux in the comparison as the analysis is closely linked to ET.

For SHF we observe a general decrease (more positive flux) by about 10 to 20 % on average per year. Summer and spring seasons exhibit different behaviours (Fig. 6(a,b)). Summer is affected by a general decrease, whereas in spring, a non-significant (p-value > 0.05) decrease is observed in forested regions while the remaining land has a greater decrease. In winter and autumn, no significant changes are happening.

Figure 5. (a) Boxplot of the differences in simulated normalized daily evapotranspiration (ET) between  $MAR_{ref}$  and  $MAR_{MODIS_{clim}}$  over the 2015–2021 period. (b) Boxplot of the differences in simulated normalized daily mean surface air temperature (TT) over the same period.

Normalization was performed seasonally, using  $MAR_{ref}$  as the reference. Boxes represent the interquartile range (25th–75th percentiles), the central line of the boxes indicates the median, and whiskers extend to 1.5 times the interquartile range. Outliers are shown as individual points.

Figure 6. Average daily difference in sensible heat flux in Belgium between  $MAR_{MODISclim}$  (using the MODIS LAI climatology) and  $MAR_{ref}$  (using the MERRA2 LAI climatology) (a) Difference in sensible heat flux in spring (b) Difference in sensible heat flux in summer. Positive difference means an increase in  $MAR_{MODISclim}$ . Forest regions are indicated with black-dashed outlines.






The changes in LWU and SWU are notably different, despite having similar variations in absolute values. LWU is less affected compared to SWU because the long-wave upwelling radiation is driven primarily by surface temperature, which remains mostly unchanged in this study (often less than 1 °C. In contrast, SWU is more influenced by changes in vegetation, as the alterations in albedo and surface reflectivity are more pronounced in these simulations. However, depending on the vegetation type, denser vegetation could lead to darker region and higher emissivity.

LWU changes are not significant in both winter and autumn (p-values > 0.05) over Belgium, with differences mainly occurring in the form of small spatial pattern changes between the simulations. During spring, forested areas experience almost no changes while the rest of the country shows a decrease in the flux. In summer, a decrease in LWU is observed across the entire country, except for coastal regions. The difference in behaviour near the coastline is explained by the higher concentration of sand in the soil composition, with an higher soil water retention capabilities.

For SWU, an increase of 5 to 10 % is observed on average, with the greater changes occurring in summer and spring. As for LWU, in winter only changes in spatial patterns are happening. These changes have no effect on the overall quantity of energy. Unlike LWU, the difference between forested regions and the other part of the study zone is less clear.

Overall, the results agree with those reported by Heck et al. (2001), whose sensitivity experiments increasing vegetation density across Europe demonstrated similar outcomes.

## 350 3.3 Sensitivity of the model to noise in LAI

Multiple simulations were then performed using the same baseline (MODIS LAI climatology), but perturbed with Gaussian noise to analyse the effect of LAI perturbations on the various climate variables used previously. The changes induced by the noise are presented in Fig. 7. A more detailed table of the changes can be found in appendix A1. As LAI is the only input that MAR takes to represent the seasonal variability of vegetation, the sensitivity to LAI can be generalized to the sensitivity of the model to a change in vegetation.

At first, we observed that the relationship between LAI and the variables is complex though most variables consistently respond either directly or inversely to changes in LAI. For example, as expected, some variables increase when LAI increases and decrease when LAI decreases, while others exhibit the opposite pattern. However, these sensitivity tests reveal that the response to perturbing LAI is non-linear for most variables, and the magnitude of the response is strongly asymmetric. Rainfall remains an exception, displaying a more indirect relationship with LAI, as discussed earlier.

The largest changes were observed in the  $MAR_{-2\mu}$  simulation, which may explain the model instability. This scenario, where LAI was reduced by an average of 92 % daily, resulted in significant drops in both evaporation and evapotranspiration (by 88.9 % and 83.4 %, respectively), alongside a notable 17.12 % increase in soil moisture content. On the contrary, despite an average daily increase in LAI of 178.4 %, the  $MAR_{+2\mu}$  simulation did not exhibit similarly extreme shifts; its changes


**Figure 7.** Average daily change (%) in surface air temperature, albedo soil moisture content, and evapotranspiration given the average change in LAI (%). None of the variables experience a linear change given the LAI when the change is statistically significant. A more detailed table of the change in variables given the change in LAI can be found in appendix A (Table A1).

were more comparable to those seen in the  $MAR_{-\mu}$  simulation.

It is also important to note that Fig. 7 does not fully reflect the seasonal sensitivity of these variables. As highlighted in Section 3.2, the evolution of certain variables varies between seasons. For example, although the overall changes in air temperature or albedo may appear small, or are not significant, these variables show stronger seasonal shifts, often with contrasting trends (positive or negative) depending on the season. Thus if the trend is positive for half of the year and negative for the remaining seasons, the yearly average is lowered. Additionally, the maximum changes do not always align with peak vegetation growth (March to June, according to MODIS LAI), but can occur before or after. For instance, the largest changes in relative humidity, air temperature, and rainfall occur in spring, while the largest changes in evapotranspiration and albedo, are observed in summer and winter.

As shown in Fig. 7, simulations with higher LAI also exhibit higher albedo and vice versa. Interestingly, this relationship appears to be non-linear; simulations with reduced vegetation experience a greater decrease in albedo than the increase observed in simulations with more vegetation.

Simulations with higher LAI tend to reduce soil moisture, while those with lower vegetation cover show an increase in soil moisture. For instance, increases of up to 21 % were observed in  $MAR_{m2m}$ , whereas  $MAR_{2m}$  exhibited decreases of less than 5 % at most. This variation is also dictated by the root water uptake, the more vegetation, the more the soil water is absorbed

**Figure 8.** (a) Distribution of summer rainfall events by hourly intensity for the different simulations between 2015 and 2021. Simulations with lower LAI tend to exhibit fewer extreme rainfall events, as these events generally occur with reduced intensity. (b) Distribution of mean daily summer surface air temperatures for the same simulations. Lower LAI simulations show a shift toward warmer conditions, with fewer days below 20°C and more days in the 20–28°C range, although the frequency of very hot days (>28°C) remains relatively unchanged.

by the plant.

by the plant





Ultimately, the overall response of each variable to changes in LAI is highly variable, and is not generalized over such long period of analysis. The intricate and sometimes contrasting seasonal patterns highlight the complexity of the interactions between vegetation and climate variables in these simulations. Therefore, a closer examination of extreme events could provide valuable insights, and will be assessed in section 3.5.

Examining the frequency of rainfall events based on their hourly intensity across the different noisy experiments,  $MAR_{ref}$ , and  $MAR_{MODIS_{clim}}$ , reveals that changes in LAI influence the distribution of rainfall events (Fig. 8(a)). While lower-intensity rainfall events show little to no variation, some of the most intense rainfall events observed in  $MAR_{MODIS_{clim}}$  are absent or appear as weaker events in  $MAR_{-\mu}$ .

Notable changes in surface air temperature distribution are also happening. While the maximum daily air temperature remains largely unaffected, the most significant impact is observed in the distribution of mean daily air temperatures during summer. Simulations with higher LAI tend to experience more days with lower air temperatures (<20 °C) and fewer days within the moderate air temperature range (20–28 °C). However, the occurrence of high air temperature days (>28 °C) does not necessarily decrease (Fig. 8(b)).




It may suggest that while an increase in LAI leads to overall cooling effects, it does not completely mitigate extreme heat events. The additional vegetation cover enhances evapotranspiration, which contributes to local cooling, particularly during moderate air temperature days. However, during extreme heat conditions, other factors such as the soil moisture availability may override the cooling influence of vegetation Vogel et al. (2017); Seneviratne et al. (2010).

# 405 3.4 Sensitivity to the integration method of the LAI

The sensitivity of MAR to the integration method of the LAI is tested through a comparison of  $MAR_{sector}$  and  $MAR_{MODIS_{clim}}$ . This comparison allows us to test for the addition of the sub-pixels of MAR in terms of vegetation. By computing the vegetation-related variable given the vegetation type and the fraction of the pixel covered by this vegetation,  $MAR_{sector}$  should have a more detailed representation of the vegetation.

For almost all the studied periods there are almost no significant changes between  $MAR_{sector}$  and  $MAR_{MODIS_{clim}}$ . However, during specific periods, there is a divergence: the only significant differences (p-values 




**Table 4.** Difference (%) in average LAI between  $MAR_{MODIS_{clim}}$  LAI values and  $MAR_{MODIS}$  in July - August for multiple years. Differences are given over the whole region of interest and considering the vegetation type (forest and low vegetation). An asterisk (\*) signifies that there is no significant difference (p-value > 0.05) between the two datasets.

| $\Delta\%$ LAI | 2018  | 2019  | 2021 | 2022  | 2023 | 2024 |
|----------------|-------|-------|------|-------|------|------|
| Belgium        | -13.2 | -5.3  | 7.6  | -10.8 | 2.2* | 6.3  |
| Forests        | -2.2* | -0.8* | 1*   | -1.3* | 0.6* | 2.1* |
| Low vegetation | -9.1  | -3.4  | 5.2  | -7.9  | 1.3* | 3.7  |

pastures (Wang et al., 2003). Therefore, the difference between  $MAR_{MODIS_{clim}}$  and  $MAR_{MODIS}$  varies depending on the regional landscape.

As shown in Fig. 9(a), in the summer 2018 forests in southern Belgium exhibited higher LAI values in  $MAR_{MODIS}$  compared to  $MAR_{MODIS_{clim}}$ . However, this trend is not consistent throughout the drought, as demonstrated in Fig. 9(c). LAI for forests initially remains higher than the climatological average but drops below by the end of July. This decline is also clearly observed for low vegetation types, such as grasslands, and can be observed more broadly across Belgium (Fig. 9(d)). This evolution is also observed during the 2019 and 2022 summer as shown in appendix B. Roughly the opposite scenario is happening during 2021, 2023, and 2024. However, for the 6 events, the average change in LAI is not statistically significant in forests and is not significant in any of the vegetation in 2023 (Table 4).

During extreme events such as those studied, the use of the 8-day LAI instead of the climatology brought roughly the same evolutions as the one in the previous sections. When the LAI in  $MAR_{MODIS}$  is higher (lower) than the one of  $MAR_{MODIS_{clim}}$  (i.e. during the pluvial (dryer) periods) the first simulation simulates more (less) evapotranspiration (Fig. 10 (a)), directly linked with the vegetation quantity. However, the difference in LAI is too small for a too short period to induce a statistically significant impact on the daily mean surface air temperature. The impact on the air temperature has to be found in the maximum surface air temperature for the day. During the pluvial periods, the larger quantity of vegetation will cool down the higher air temperature and the lower quantity of vegetation during droughts will increase them. This effect is stronger for the dryer period and for the days where the maximum air temperature is moderately high (around 20 to 30 °C), as already observed in the sensitivity tests (Fig. 10 (b)). However, even though differences are visible between the two models, none of them are statistically significant neither in forests nor in low vegetation when comparing over both July and August. Differences became statistically significant when using smaller time windows of 3 to 5 days. However, it is important to note that while a 0.5 to 1 degree of difference in the daily maximum air temperature may not be statistically significant, having 0.5 °C of difference on average over the whole summer could have a significant impact on ecosystems (Kaufmann et al., 2003; Li and Convertino, 2021).

In addition, the timing of maximum differences in ET does not consistently occur during the same period of summer across the six years studied, nor does it align between vegetation types. For instance, the highest 7-day average differences were

Figure 9. Difference in LAI between the MODIS LAI climatology (used in  $MAR_{MODIS_{clim}}$ ) and the 8-day MODIS LAI data (used in  $MAR_{MODIS}$ ) in July and August 2018. (a) Map of the average difference (in %) in Belgium. Forest regions are indicated with the black lines. (b) Average difference over Belgium. (c) Average difference for the forests. (d) Average difference for low grass. The pink line represents the 8-day MODIS LAI data and the blue line represents MODIS LAI climatology.

Figure 10. Daily difference between  $MAR_{MODIS}$  and  $MAR_{MODIS_{clim}}$  over Belgium during summer 2018, 2019, 2021, 2022, 2023, and 2024. (a) Mean daily evapotranspiration for the two model. The scatter plot has been cut between 1 and 4 mm. (b) Maximum daily surface air temperature for the two model. The scatter plot has been cut between 25 and 35 °C.

observed in early August for both forest and low vegetation in 2018 and 2019. In 2021, the peak occurred in mid-June for low vegetation and mid-July for forest. In contrast, it appeared in late March 2022, early September 2023, and late July 2024.

### 4 Discussion



The sensitivity of MAR meteorological variables to its LAI input is non-linear and requires an in-depth analysis of the results to fully understand the impact of altering the vegetation parameters. Replacing the MERRA2-based LAI with the MODIS LAI a led to significant changes in the model outputs and generally improved the overall performance because of the higher spatial resolution. In contrast, incorporating the 8-day MODIS LAI dataset, while offering a dynamic LAI compared to s static one, resulted in comparatively minor changes, even during extreme events. Focusing on the summer periods (July–August) of 2018, 2019, 2021, 2022, and 2023, the use of MODIS LAI instead of MERRA2 LAI improved the simulation of both maximum daily air temperature (Fig. 11a) and daily average evapotranspiration (Fig. 11b). In most cases, the correlation coefficient between the model outputs and observations increased, and the model's standard deviation approached that of the observed values. Although significant differences were observed for both variables, the improvement was more pronounced for ET. However, the improvement of using the 8-day LAI is less pronounced as is some cases, the model get further from the observations.


Figure 11. Taylor diagram for  $MAR_{ref}$ ,  $MAR_{MODIS}$ , and  $MAR_{MODIS_{clim}}$  for July and August of drought (2018 ( $\bullet$ ), 2019 (+) and, 2022 ( $\blacklozenge$ )), and pluvial years (2021 ( $\blacktriangledown$ ), and 2023 ( $\blacktriangle$ )). (a) Taylor diagram for average daily evapotranspiration (ET) (b) Taylor diagram for daily maximum surface air temperature. Points of the same colour are values from the same the model while point of the same shape represent the same summer.

These findings suggest that MAR is sensitive to its LAI inputs; however, the coupling between the atmosphere and land surface may appears relatively too resilient in comparison to observed coupling from these regions (Knist et al., 2017).

The results are consistent with previous studies investigating scenarios with increased vegetation at regional or global scales, particularly regarding changes in soil moisture, evaporation, ET, surface air temperature, and albedo (e.g., Jiménez-Gutiérrez et al. (2021); Engel et al. (2025); King et al. (2024)). However, unlike some of those studies, the current experiments did not reproduce the associated changes in rainfall. Specifically, the observed increase in ET did not result in a corresponding increase in total or convective rainfall, contrary to expectations based on typical regional feedback mechanisms (van der Ent et al., 2010; Eltahir and Bras, 1996).

Moreover, the use of near-real-time observed LAI did not always lead to statistically significant improvements of the model performance, even during extreme events that visibly impacted vegetation cover. This limited sensitivity may reflect the resilience of the ecosystems within the study region, where the effects of climate change and extreme events tend to emerge over longer timescales, such as through increased forest mortality (Knutzen et al., 2025; Schuldt et al., 2020), or a too conservative model. Nonetheless, the decrease in extreme rainfall in the simulations with less vegetation (Fig. 8) is extremely interesting, it may be the sign that the model may not correctly represents extreme events in the future.

These outcomes underscore the importance of integrating a more detailed and dynamic representation of vegetation, particularly given the uncertainties surrounding future vegetation states. As tree cover and land use are expected to evolve significantly in the coming decades (Roebroek et al., 2025), relying on a fixed climatological LAI is likely insufficient to fully capture future land–atmosphere interactions.

However, the use of remote sensing data in these experiments comes with limitations. While the 8-day MODIS LAI product is valuable for analysing past events and seasonal variability, it does not support future climate projections. A promising pathway to address these limitations lies in dynamic model coupling (Giorgi and Gao, 2018). Coupling MAR with a dynamic vegetation model such as the CARbon Assimilation in the Biosphere (CARAIB; Dury et al., 2011) model would provide temporally evolving and more physiologically realistic vegetation inputs for both present and future simulations. In addition to LAI, this approach would allow the incorporation of a broader range of vegetation-related variables, enhancing the representation of land–atmosphere interactions and potentially helping to reduce model uncertainties (Krinner et al., 2005).

#### 5 Conclusions



In this work, we investigated the representation of biosphere-atmosphere interactions in the MAR model and quantify the sensitivity of some of its meteorological variables by using remote sensing-based LAI dataset.

Overall, the comparison between the model's current configuration  $(MAR_{ref})$  and the version using MODIS LAI climatology  $(MAR_{MODIS_{clim}})$  highlights the critical importance of accurate vegetation representation in climate modelling. The results suggest that the current MAR configuration, which relies on MERRA2 LAI, may underestimate the vegetation variability and density in Belgium. This underestimation could contribute to a spurious drying trend over multiple years, potentially compromising the model's reliably to accurately project climate variables over extended periods.

The results from our sensitivity analyses demonstrate that changes in vegetation inputs can significantly impact the outputs of the MAR model. One of the most pronounced impacts of altering LAI was on ET, with  $MAR_{MODIS_{clim}}$  consistently producing higher cumulative ET than the reference MAR run. This result aligns with the generally higher LAI values in  $MAR_{MODIS_{clim}}$  compared to the reference, which led to more vegetation activity and consequently greater water uptake and release through transpiration. This effect was particularly pronounced during the spring/summer months, as expected.

Surface air temperature responses to LAI changes also varied seasonally. In summer, simulations with higher LAI resulted in cooler temperatures. In contrast, during winter, these simulations exhibited slightly warmer air temperatures, a phenomenon that may be attributed to the insulating effect of vegetation and reduced heat loss. The opposite conclusion can be drawn for simulation with lower LAI.

The comparison between the unperturbed simulations and the simulation with added noise highlighted that some variables, were almost not impacted by the change in LAI input; e.g., Despite significant changes in LAI, rainfall only exhibits minor





variations, suggesting that the link vegetation and rainfall may be highly indirect.

Another key finding is the non-symmetrical effect of the LAI change. The different experiments pinpointed that there is no linear link between the change in LAI the studied variables: e.g., while the simulation using the MODIS climatology showed a moderate decrease in soil moisture, simulations with even more dense vegetation experienced only a slightly bigger decrease. Conversely, simulations with lower vegetation cover showed a substantial increase in soil moisture, up to 11 % in summer on average. This variation is primarily driven by changes in root water uptake, which intensifies with denser vegetation and reduces as vegetation becomes sparser. The same effect is occurring in surface air temperature, where doubling the LAI does not result in a linear response; instead, it can lead to nearly a threefold increase. This air temperature response highlights the complex interplay between vegetation, surface albedo, and energy fluxes, demonstrating the retro-action of vegetation to the atmosphere.

The sensitivity experiment also demonstrated the seasonality effect. On surface air temperature and relative humidity, where the most change is happening during spring during the vegetation peak, and on evapotranspiration where it happens in summer or winter.

In addition, the role of short-term dynamic vegetation (7-days) has been highlighted by using the MODIS 8-day LAI. Although, while surprisingly the differences with the reference were lower than not using a climatology during drought, it led to changes in LAI that were different given the vegetation type. These changes impacted the model results spatially and temporally.

Under the current climate, the results indicate that low vegetation, compared to forests, was the most affected during droughts. This was linked to a more significant decrease in LAI in these types of vegetation than in forests. Special attention should be given to low vegetation or vegetation more susceptible to water stress in RCMs, as variations were more pronounced. However, the forests should not be neglected. Extreme events are predicted to be more frequent and more strong (on Climate Change, IPCC) and forest will suffer even more in the future (Knutzen et al., 2025). RCM will required to correctly represent forest for precise previsions and should account for long term dynamics.

When compared to observations, the model correctly represents the climate over a longer time span but struggles to replicate some specific events that occur over a shorter period. The use of an observation-based LAI dataset allowed to better represent some of these events such as the 2018 drought. However, the impact of using the 8-day LAI was less clear than changing the dataset, pointing toward limitations in the model representation of the land-atmosphere interactions.

These findings underscore the crucial role that vegetation plays in RCMs. While the MAR model can currently be adjusted or forced with observational data to better reflect the existing climate, relying on a static LAI climatology to represent vegetation could lead to inaccurate representations of the future climate and extreme events or even the impossibility to represent them. The importance of this issue becomes even more pronounced when considering future climate scenarios, where uncertainties

regarding vegetation states may increase.

Future research should focus on refining vegetation inputs in MAR and other regional climate models. This refining can be performed through the coupling of DVMs with RCMs. Incorporating dynamic vegetation represents a critical advancement in climate modelling (Gröger et al., 2021; Giorgi and Gao, 2018). While such integration has already been implemented in some RCMs (Zhang et al., 2014) and explored through sensitivity experiments (Heck et al., 2001; Jiménez-Gutiérrez et al., 2021), it remains far from standard practice. Given that our findings show vegetation changes already have a significant impact on the present climate, this step is essential for reducing uncertainties and to ensure data continuity in future climate projections.

Code and data availability. The MAR code used in this study is tagged as v3.13 on https://gitlab.com/Mar-Group/MARv3 (MAR model, 2025). The MAR outputs used in this study, Python code, and the necessary files to perform the perturbation to MODIS LAI are available upon request by email (tdethinne@uliege.be).

# Appendix A: Evolution of variable compared to the noise

Table A1. Mean daily difference (%) between  $MAR_{MODIS_{clim}}$  (using the MODIS LAI climatology) and the simulations with Gaussian noise applied to their LAI from 2015 to 2021 (excepted for  $MAR_{-2\mu}$  2015-2019). Refer to Table 1 for the description and name of the simulations. An asterisk (\*) signifies that there is no significant difference (p-value > 0.05) between  $MAR_{MODIS_{clim}}$  daily values and the perturbed experiments.

| Variable              | $MAR_{+2\mu}$          | $MAR_{+\mu}$          | $MAR_{+0}$          | $MAR_{-0}$           | $MAR_{-\mu}$           | $MAR_{-2\mu}$          |
|-----------------------|------------------------|-----------------------|---------------------|----------------------|------------------------|------------------------|
| Noise                 | Gaus. $(2\mu, \sigma)$ | Gaus. $(\mu, \sigma)$ | Gaus. $(0, \sigma)$ | -Gaus. $(0, \sigma)$ | -Gaus. $(\mu, \sigma)$ | -Gaus. $(2\mu,\sigma)$ |
| Leaf Area Index       | 178.4                  | 81.0                  | 16.6                | -33.1                | -58.8                  | -92.0                  |
| Temperature           | -0.54*                 | -0.53*                | -0.15*              | 0.61*                | 0.84*                  | 0.61*                  |
| Albedo                | 3.59                   | 1.42                  | -1.63               | -1.47                | -8.1                   | -16.13                 |
| Relative humidity     | 2.23                   | 1.56                  | 0.35*               | -0.88*               | -2.95                  | -4.97                  |
| Soil moisture content | -4.19                  | -3.78                 | -1.23               | 2.41                 | 9.35                   | 17.12                  |
| Rainfall              | 8.62*                  | 5.71*                 | 1.33*               | -1.63*               | 4.74*                  | 9.72*                  |
| Evapotranspiration    | 29.79                  | 16.44                 | 4.6*                | -4.91*               | -30.14                 | -83.4                  |
| Evaporation           | 27.36                  | 14.99                 | -0.27*              | -14                  | -35.24                 | -88.9                  |
| Latent Heat Flux      | 4.98                   | 5.46                  | 1.83*               | -2.97                | -14.04                 | -30.5                  |
| Sensible Heat Flux    | -14.09                 | -11.2                 | -1.75*              | 8.16                 | 7.83                   | 8.5                    |

# Appendix B: Difference in LAI in summer

Figure B1. Map of the average difference (in %) in LAI between the MODIS LAI climatology (used in  $MAR_{MODIS_{clim}}$ ) and the 8-day MODIS LAI data (used in  $MAR_{MODIS}$ ) in July and August 2019 in Belgium.

Figure B2. Map of the average difference (in %) in LAI between the MODIS LAI climatology (used in  $MAR_{MODIS_{clim}}$ ) and the 8-day MODIS LAI data (used in  $MAR_{MODIS}$ ) in July and August 2020 in Belgium.

Figure B3. Map of the average difference (in %) in LAI between the MODIS LAI climatology (used in  $MAR_{MODIS_{clim}}$ ) and the 8-day MODIS LAI data (used in  $MAR_{MODIS}$ ) in July and August 2021 in Belgium.

Figure B4. Map of the average difference (in %) in LAI between the MODIS LAI climatology (used in  $MAR_{MODIS_{clim}}$ ) and the 8-day MODIS LAI data (used in  $MAR_{MODIS}$ ) in July and August 2022 in Belgium.

Figure B5. Map of the average difference (in %) in LAI between the MODIS LAI climatology (used in  $MAR_{MODIS_{clim}}$ ) and the 8-day MODIS LAI data (used in  $MAR_{MODIS}$ ) in July and August 2023 in Belgium.



Figure B6. Map of the average difference (in %) in LAI between the MODIS LAI climatology (used in  $MAR_{MODIS_{clim}}$ ) and the 8-day MODIS LAI data (used in  $MAR_{MODIS}$ ) in July and August 2024 in Belgium.

Author contributions. T. D., N. G., X. F., and F. J. conceived the study. T. D. performed the simulations. T. D. processed the remote sensing and model data. N. G. and C. K. assisted with evaluation of the model. T. D. led the writing of the manuscript. T. D., N. G., C. K., B. L., X. F., and F. J. discussed the results. All co-authors revised and contributed to the editing of the manuscript.

Competing interests. The authors declare that they have no conflict of interest.

Acknowledgements. ERA5 reanalysis data (Hersbach et al., 2020) are provided by the European Centre for Medium-Range Weather Forecasts, from their website at https://www.ecmwf.int/en/forecasts/datasets/reanalysis-datasets/era5 (last access: 24 October 2022). MET/DMET [LSA-311; LSA-312] was provided by the EUMETSAT Satellite Application Facility on Land Surface Analysis (LSA; Trigo et al., 2011) The dataset of observation gridded over Belgium was provider by the Royal institute of meteorology of Belgium. The authors acknowledge the Consortium des Équipements de Calcul Intensif (CÉCI), funded by the Fonds de la Recherche Scientifique de Belgique (F.R.S. – FNRS) under grant no. 2.5020.11 and the Tier-1 supercomputer (Nic5) of the Fédération Wallonie Bruxelles infrastructure funded by the Walloon Region under grant agreement no. 1117545. The authors also thanks to Baudewyn Miriam for the SPEI3 computation. N.G. acknowledge support by the Belspo-Fed-twin. C. K. acknowledge support by the Fonds de la Recherche Scientifique – FNRS. B. L. acknowledges support by the Fonds pour la Formation à la Recherche dans l'Industrie et dans l'Agriculture (FRIA, Belgium). The publication's spelling and grammar have been thoroughly reviewed with the assistance of AI-powered tools.

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
