# Peer review of "Assessing Regional Climate Model Sensitivity to Vegetation Dynamics Informed by Remote Sensing"

_EGUsphere, 2025_

## Author Comment (AC1)

**Reply on RC1**

**General comments**

Although the article does not introduce new scientific methods, the methods used are fully sufficient for a detailed analysis of the specific phenomenon under investigation.

The language and the interpretation of the results are clear, and it shows very well the influence of LAI on atmospheric parameters produced by MAR. In addition to the objectives of the article, it also clearly shows the role of forests in the natural environment.

Nevertheless, here are some points for possible amendments and explanation.

-the title does not fully reflect the content of the manuscript as the vegetation dynamic is represented exclusively only by one parameter - LAI. Therefore, LAI should be reflected in the title. Further to that, the role of LAI in vegetation dynamics should be briefly discussed in text.

Thank you for this relevant comment. We agree that vegetation dynamics in our study are represented exclusively through LAI, which is a central aspect of the manuscript. Accordingly, we propose the following revised title:

  **"Assessing Regional Climate Model Sensitivity to Vegetation Dynamics Using Spaceborne Remote Sensing-derived Leaf Area Index."**

In addition, we will include a short paragraph in the LAI data section clarifying the role of LAI in representing vegetation dynamics

Line 174 will be changed to : "As explained in Section 2.1, the vegetation cycle in MAR is externally prescribed. In this case, MAR uses as primary information the Leaf Area Index (LAI), that represents the one-sided green leaf area of a canopy per unit ground area, influencing processes like photosynthesis, and energy exchange at the vegetation-atmosphere interface  (Chen and Black, 1992).  Because carbon and water exchanges between vegetation and the atmosphere occur at the leaf scale, LAI is closely linked to land-atmosphere fluxes and is therefore a key variable in climate modelling (Rogers et al., 2021). In addition, LAI is strongly associated with major phenological stages such as leaf expansion, maturity, and senescence, making it a direct indicator of seasonal vegetation dynamics (Richardson et al., 2013)."

-the finding that increased LAI leads to increased evaporation is somehow surprising. Does it have any physical explanation, or could it rather be attributed to MAR procedure?

The increase in evaporation is indeed induced by the MAR physics. The evaporation in MAR combines the evaporation of open water, water in the first centimeters of soil and water intercepted by the leaves. As MAR tends to quickly make the water drain below

the modelled soil depth, the water reaching the ground tends to reach depth at which it is no longer considered for the surface soil evaporation. Thus by increasing the LAI, more water is intercepted by the leaves and thus more water is evaporated.

Definitions of evaporation and evapotranspiration will be provided in the text.

-it should be clarified what data entered the comparison in Sec. 3.1. The terms „observational-modelled data" (row 255) and „observed data" (row 259) should be defined/described.

Thank you for this comment. "Observed data" in line 259 is indeed a typo.

Line 255 will be revised to:
 "To evaluate the simulations, we compared the daily averaged model outputs against datasets directly derived from observations."

Line 259 will be revised to:
 "[…] perform a statistical comparison between the model and observation-based data."

In this manuscript, we define observation-based data as data derived directly from observations that have undergone post-processing prior to their use in the analysis.

-the sensitivity test in section  3.4 should be supported by some numeric or graphic interpretation.

Thank you for this suggestion. We did not initially include numerical or graphical interpretations of the sensitivity test results because the differences were not statistically significant. However, to improve clarity and completeness, we will include an additional table in appendix (Table C1), similar to Table 3, presenting seasonal daily average values of surface air temperature, soil moisture content, and albedo, as well as cumulative mean daily rainfall and evapotranspiration for summer (JJA) and winter (DJF) over the 2015–2024 period.

Table C1 : Daily average values of surface air temperature, soil moisture content, and albedo, along with cumulative daily rainfall and evapotranspiration, for summer (JJA) and winter (DJF) during 2015–2024, as simulated by $MAR_{MODISclim}$ (using the MODIS LAI climatology) and $MAR_{sector}$ (using the MODIS LAI climatology corrected for MAR subpixel).

| | Temperature (°C) | | Soil moisture | | Albedo | | Rainfall (mm/day) | | ET (mm/day) | |
|---|---|---|---|---|---|---|---|---|---|---|
| | DJF | JJA | DJF | JJA | DJF | JJA | DJF | JJA | DJF | JJA |
| $MAR_{MO}$ | 3.8 | 18.4 | 0.65 | 0.63 | 26.3 | 23.8 | 194 | 232 | 6 | 156 |

| | | | | | | | | | | |
|---|---|---|---|---|---|---|---|---|---|---|
| DIS
MAR$_{sect}$
or | 3.8 | 18.6 | 0.65 | 0.63 | 26.2 | 23.5 | 190 | 227 | 6 | 158 |

-the statement „...that changes in LAI influence the distribution of rainfall events (Fig. 8(a)) sounds like LAI is influencing the raifall events didtribution.The above fact is rather attributed to the seasonal distribution of both LAI and rainfall events. Explain please, if not.

Thank you for pointing out this ambiguity. We agree that the seasonal co-variation of LAI and rainfall should not be interpreted as a causal influence of LAI on the temporal distribution of rainfall events. In our simulations, changes in LAI do not modify the temporal distribution of rainfall events, but only the magnitude of some extreme rainfall enhanced by local (inside MAR integration domain) evapotranspiration.

To avoid any confusion, lines 391–392 will be revised as follows:
"Examining the frequency of rainfall events based on their hourly intensity across the different experiments (MAR$_{+-x\mu}$, MAR$_{ref}$, and MAR$_{MODISclim}$) reveals that changes in LAI influence only the magnitude of some high-intensity rainfall events (Fig. 8(a))."

**Typos**

There are very few typos in the text. Nevertheless, one more check for typos, like in rows 31 and 465 is recommended.

The authors would like to thank Dr Nejedlik Pavol for identifying typographical errors. The text will be reviewed and the typos will be changed accordingly.

-row 31  . n response, ... (In response,...)

-row 465 ...a led to... (skip a),   ...LAI compared to s static one,... (change to s to a)

**Formal notices**

-the areas marked with a thin dashed line in Fig. 1 should be defined in the legend to Fig. 1.

Thank you for this remark. The areas marked with dashed lines correspond to forested regions within the studied domain. The figure 1 caption will be revised to include the following sentence:

"Forested (or partially forested) regions of Ardenne and Campines are indicated with black dashed outlines."

-it should be stressed that 10 years of phenological data is quite a short time to compute useful climatology (row 193). The word „averages" instead of „climatology" should be rather used.

Thank you for this important remark. We agree that a 10-year period is relatively short to establish a true climatology. Accordingly, the term "climatology" will be replaced by "average" throughout the manuscript when referring to the MODIS dataset. For example, line 192 will be revised to:

"From the 8-day MODIS product, we computed an annual daily average using all available images over the period 2012–2022."

$MAR_{MODISclim}$ will also be renamed $MAR_{MODISavg}$

-it is difficult to distinguish some of the lines in Fig, 4a.

We agree that $MAR_{MODISsector}$ and $MAR_{-0}$ are difficult to distinguish. The colour palette will be revised to improve visual contrast and readability.

-terms $MAR_{m2m}$ and $MAR_{2m}$ (row 382) should be clearly defined.

We thank the reviewer for noting the inconsistent terminology. The terms $MAR_{m2m}$, $MAR_{2m}$, and $MAR_{MERRAclim}$ correspond to earlier naming conventions and will be replaced by the experiment names used in Table 1, namely $MAR_{-2\mu}$, $MAR_{+2\mu}$, and $MAR_{ref}$, respectively. The entire manuscript will be carefully reviewed to ensure consistency between text, figures, and tables.

-rows 468 and 469. ... maximum daily air temperature (Fig. 11a) and daily average evapotranspiration (Fig. 11b). Figs 11a and 11b show the ET and Tmax in reverse order from that listed in the text.

We thank the reviewer for identifying the mismatch between the text and the figure order. The text will be corrected so that maximum daily air temperature (Fig. 11b) and daily average evapotranspiration (Fig. 11a) are described consistently with the figure. The text will be reviewed to ensure consistency between the text and the figure/tables.

-Fig. 11. The description in the legend and in the figure itself should be identical. $MAR_{MERRAclim}$ is shown in the figure while $MAR_{ref}$ in the legend.

Following precedent comments, $MAR_{MERRAclim}$ corresponds to earlier naming conventions and will be replaced by the experiment names used in Table 1 namely $MAR_{ref.}$

-row 353. There is no Appendix A1. There is Appendix A, Table1.

The reference to Appendix A1 is incorrect and the line 353 will be changed to "A more detailed table of the changes can be found in table A1 in appendix A"

**Bibliography**

- Chen, J. M. and Black, T. A.: Defining leaf area index for non-flat leaves, Plant, Cell & Environment, 15, 421–429, https://doi.org/10.1111/j.1365-3040.1992.tb00992.x, 1992.

- Richardson, A. D., Keenan, T. F., Migliavacca, M., Ryu, Y., Sonnentag, O., and Toomey, M.: Climate change, phenology, and phenological control of vegetation feedbacks to the climate system, Agricultural and Forest Meteorology, 169, 156–173, https://doi.org/10.1016/j.agrformet.2012.09.012, 2013.

- Rogers, C., Chen, J. M., Croft, H., Gonsamo, A., Luo, X., Bartlett, P., and Staebler, R. M.: Daily leaf area index from photosynthetically active radiation for long term records of canopy structure and leaf phenology, Agricultural and Forest Meteorology, 304-305, 108 407, https://doi.org/10.1016/j.agrformet.2021.108407, 2021.

---

## Author Comment (AC2)

**Reply on RC2**

This preprint presents a multi-experiment sensitivity study of the MAR regional climate model (SISVAT surface scheme) to vegetation forcing by Leaf Area Index (LAI), replacing the default MERRA2-based LAI climatology with MODIS LAI (climatology and 8-day) and adding synthetic LAI perturbations, with the main finding of strong, nonlinear/asymmetric impacts on ET/evaporation/soil moisture and weaker effects on temperature and especially precipitation; however, several core methodological and reporting issues (notably unit/definition inconsistencies, confounding changes between baseline experiments, incomplete specification of the LAI perturbation procedure, and significance testing that likely ignores autocorrelation/spatial dependence) currently prevent unambiguous attribution of the simulated differences to vegetation dynamics and weaken the robustness of the conclusions.

**Major comments:**

1. The manuscript contains critical internal inconsistencies in units/definitions that must be resolved before results can be interpreted: Table 3 reports summer ET/rainfall values (e.g., 120–156 "mm" ET; 230–232 "mm" rainfall) while text refers to "mean daily"/"cumulative mean daily" in ways that imply contradictory units (mm/day vs mm/season); soil moisture is reported as ~65–68% in Table 3 but described as 0.72%→0.56% in text (likely fractions mis-labeled as percent); "evaporation" vs "evapotranspiration" is not defined (soil evaporation only vs soil+interception, etc.), which is essential for interpreting large asymmetries.

Thank you for highlighting these important inconsistencies. We agree that unit and definition clarity is essential for correct interpretation of the results.

All units will be carefully reviewed and corrected throughout the manuscript to ensure consistency between the text, tables, and figures. In Table 3, evapotranspiration and rainfall values will be explicitly expressed in mm day$^{-1}$, and soil moisture will be consistently reported in fraction.

In addition, Section 3.1 will be expanded to clearly define the physical variables used in this study. In particular, we will explicitly distinguish evaporation (defined as the evaporation from soil, open water and water on the leaves) from evapotranspiration (defined as the combination of evaporation and plant transpiration) and explicitly state that the soil water content is expressed as the fraction of soil water saturation.

2. The key comparison MARref vs MARMODISclim is confounded and does not isolate spatial/temporal resolution effects: (i) MERRA2 LAI climatology is for 1961–1990 while MODIS climatology is 2012–2022 (different climate/land-management era), and (ii) MAR applies vegetation-type correction coefficients to MERRA2 LAI but not to MODIS LAI (except partially via MARsector). The reported "improvements" and sensitivity could be driven by

rescaling/calibration rather than information content/resolution. Add bridging experiments (e.g., MERRA2 without coefficients; MODIS with comparable scaling; bias/quantile matching at coarse scale) or quantify the portion of LAI differences attributable to scaling vs source.

We thank the reviewer for this important comment. We agree that the observed differences between $MAR_{ref}$ and $MAR_{MODISclim}$ are likely influenced not only by spatial resolution but also by differences in the period of record, calibration, and construction of the LAI datasets, as well as the application of vegetation-type correction coefficients in MAR.

To avoid any misleading interpretation, the title of Section 3.4 will be revised to: "Impact of the new LAI dataset", and references to "sensitivity to spatial resolution of the LAI input" will be replaced with wording that emphasizes the impact of using an alternative LAI dataset.

While bridging experiments (e.g., MERRA2 without coefficients or MODIS with comparable scaling) would be ideal, they are beyond the scope of the present study. The text will include a clear discussion acknowledging that the reported differences may arise from dataset era, calibration, and construction, rather than the spatial resolution alone.

3. The LAI "Gaussian noise" design is insufficiently specified and is closer to a large systematic bias injection than random noise: setting the noise mean to ±μ or ±2μ of daily mean LAI creates strong deterministic shifts, and missing details about (a) clipping/capping of LAI (0 bound and realistic maxima), (b) frequency/location of negative values and how handled, and (c) spatial/temporal correlation structure (pixel-wise white noise vs coherent field vs correlated random field) make the resulting nonlinearity/asymmetry difficult to trust. Report the full perturbation formulation, bounds, clipping rates, and resulting LAI distributions by land-cover class and season.

We thank the reviewer for this important remark. The LAI perturbation is indeed better described as a systematic bias combined with a stochastic Gaussian variability rather than as pure random noise. We will rewrite the description accordingly and now provide the full mathematical formulation of the perturbation, the probability density function, and explicit information on bounds and clipping.

The description of the noise will be changed to : "The LAI perturbation experiments do not represent pure stochastic noise but rather a systematic bias applied to the daily LAI climatology, combined with a random Gaussian variability. For each day, the perturbation is defined as:

$LAI_{pert}(x,t) = LAI_{clim}(x,t) + μ(t) + ε(x,t)$

where $LAI_{clim}$ is the daily MODIS climatological LAI, μ(t) is a deterministic bias equal to ±μ or ±2μ of the domain-mean daily LAI, and ε(x,t) is a Gaussian random noise with zero

mean and standard deviation equal to the spatial standard deviation of LAI for that day. The probability density function of ε is given by Eq. (1).

$$p(x) = \frac{1}{\sqrt{2\pi\sigma^2}} e^{-\frac{(x-\mu)^2}{2\sigma^2}} \qquad (1)$$

where μ is the mean and σ the standard deviation. Negative LAI values were clipped to zero and no upper bound was imposed."

The perturbation was applied at the native 500m MODIS resolution prior to resampling to the MAR grid. We now report the percentage of pixels affected by clipping by season and perturbation amplitude (Table R2) and show that, although clipping is frequent at 500 m under the −2μ perturbation in spring, the aggregated 5-km LAI rarely reaches zero.

The random component is applied independently to each pixel and day, without additional imposed spatial or temporal correlation. We acknowledge that the asymmetry partly reflects the nonlinear impact of the LAI lower bound and now explicitly discuss this limitation in the manuscript.

Table R1: Percentage of pixel clipped to 0 in the MODIS images given the season and the perturbation noise applied to the LAI and the resulting number of pixels with a 0 value in MAR.

|  | -Season | +2μ | +μ | +0 | -0 | -μ | -2μ |
|---|---|---|---|---|---|---|---|
| MODIS | Winter | 0.0 | 0.0 | 0.0 | 0.3 | 6.2 | 34.5 |
|  | Spring | 0.0 | 0.0 | 0.0 | 3.9 | 15.9 | 37.1 |
|  | Summer | 0.0 | 0.0 | 0.0 | 9.9 | 17.2 | 35.5 |
|  | Autumn | 0.0 | 0.0 | 0.0 | 4.9 | 11.6 | 33.3 |
| MAR | Winter | 0 | 0 | 0 | 0 | 0 | 1 |
|  | Spring | 0 | 0 | 0 | 0 | 0 | 2.4 |
|  | Summer | 0 | 0 | 0 | 0 | 0 | 1.1 |
|  | Autumn | 0 | 0 | 0 | 0 | 0 | 0.5 |

4. The strong asymmetry (large response to LAI decrease vs smaller response to LAI increase) may be partly an artifact of hard bounds/clipping and diminishing returns; it must be demonstrated as physical rather than numerical by showing (i) whether LAI hits 0 frequently in negative experiments, (ii) whether positive experiments saturate due to parameter caps (e.g., canopy resistance, albedo formulation), and (iii) sensitivity under multiplicative bounded perturbations (LAI×(1+ε)) rather than additive bias that can force zeros.

We thank the reviewer for this insightful comment.

**(i)** As explained in our response to Comment #3, LAI reaches zero mainly in the MAR−2μ experiment and only for a few percent of the MAR pixels. This information will be explicitly recalled in the manuscript to clarify its role in the asymmetry.

**(ii)** The potential saturation of vegetation-related parameters is indeed a key element of the model sensitivity explored in this study. Although not explicitly stated in the current version, the SISVAT vegetation parameter formulations inherently control the nonlinear response of LAI variations. We will therefore add a dedicated discussion identifying which SISVAT parameters (e.g., canopy resistance, albedo, roughness length, interception rainfall storage) may reach saturation and contribute to the observed asymmetry in the model response.

**(iii)** While we understand the reviewer's concern regarding the use of additive bias perturbations, we consider that this methodology still captures the essential part of the model sensitivity to LAI. Nevertheless, we will acknowledge in the discussion that multiplicative bounded perturbations could represent an alternative approach and that future work could further investigate this aspect.

5. Statistical significance claims are likely overstated: daily meteorological/flux time series are autocorrelated, precipitation is non-normal/zero-inflated, and spatial maps involve multiple comparisons; simply applying p<0.05 on daily samples (and across pixels) without effective sample size correction, block bootstrapping, or field-significance/FDR control is not defensible. Recompute significance with autocorrelation-aware methods and address spatial multiple testing.

We thank the reviewer for this important remark. We acknowledge that the direct application of a Mann–Whitney U test on daily samples, without accounting for autocorrelation and multiple testing, represents a methodological limitation.

To address this limitation, we recomputed the statistical significance using a block bootstrap approach with 10-day blocks. This method was applied to all variables and seasons and will be specified in the manuscript.

The results show that only a limited number of variable-season combinations change their statistical significance compared to the original Mann-Whitney test. Importantly, in all cases where the significance classification changed, the corresponding daily mean bias remained smaller than the seasonal daily standard deviation, so the physical interpretation and conclusions of the study are unchanged.

We therefore retain the same conclusions but now base the statistical significance on the autocorrelation-aware block bootstrap method. The manuscript will be revised accordingly, and the methodological testing will be explicitly discussed.

For the spatial autocorrelations, using a daily value averaged over the study area should reduce its impact.

6. Interpretation of weak precipitation sensitivity must account for model configuration constraints: the domain is small and strongly forced by ERA5 (6-hour boundaries + nudging aloft), so synoptic control may dominate and suppress land-surface feedbacks on rainfall. If concluding limited rainfall sensitivity, provide process diagnostics (convective vs stratiform partition if available; low-level moisture convergence; PBLH/LCL/CAPE changes; moisture budget) to show whether ET changes could plausibly influence convection under this setup.

We thank the reviewer for this important remark. We agree that the limited precipitation sensitivity observed in our experiments must be interpreted in light of the MAR configuration, namely the relatively small integration domain size and the strong large-scale forcing imposed by ERA5 based lateral boundary conditions and upper-level nudging. Under such conditions, synoptic-scale control is expected to dominate and may substantially limit local land-surface feedback on precipitation.

In MAR, sensitivity to convection, particularly convective precipitation, is known to be weak, especially for small domains, as demonstrated by Doutreloup (2022). In our experiments, convective precipitation exhibits very small to negligible changes so it was not explicitly discussed in the current manuscript.

Unfortunately, LCL and CAPE are not available in the present experimental setup. However, we will extend the discussion section to explicitly address these limitations and to formulate hypotheses regarding the potential processes involved. In particular, we will discuss the possible roles of convective versus stratiform precipitation partitioning, low-level moisture convergence, boundary-layer height, and moisture budget constraints, and how these factors may prevent evapotranspiration changes from effectively influencing convection and precipitation under strong synoptic control.

This additional discussion will clarify that the weak precipitation response is likely a consequence of the model configuration rather than an absence of physical land-atmosphere coupling.

7. The reported albedo response direction ("higher LAI increases albedo") is not generally expected and depends on canopy optics, background soil, snow masking, and land-cover; without mechanistic evidence this will be questioned. Provide SISVAT albedo parameterization details (how LAI enters), and stratify albedo responses by land-cover and snow/no-snow conditions to justify the sign and seasonality.

In SISVAT, surface albedo is parameterized as a weighted combination of soil and vegetation albedo, with LAI controlling the relative contribution of each component as well as the absorption and scattering of direct and diffuse radiation following the formulation described in De Ridder (1997). Increasing LAI reduces the contribution of the underlying soil reflectance and increases the dominance of canopy optical properties.

In our study domain, the vegetation is dominated by low vegetation and broadleaved forest types. For both classes in SISVAT, canopy albedo is higher than the underlying soil albedo, so increasing LAI leads to a net increase in grid-cell albedo. This behaviour is therefore consistent with the SISVAT parameterization and with the vegetation composition of the domain.

Snow masking effects are expected to be limited in our seasonal averages, as only 87 days with snow cover exceeding 10% of the domain (and ≥1 cm depth) occurred in the MAR$_{MODISclim}$ simulation over the full study period. Snow therefore does not significantly affect the multi-year seasonal albedo averages discussed here.

As explained in Comment #4, we will (i) add a short description of the SISVAT albedo formulation and the role of LAI, and (ii) stratify the albedo response by vegetation class in the manuscript.

8.  The statement that LAI underestimation contributes to a multi-year "spurious drying trend" is not supported without a closed water balance diagnosis. Provide P–ET–runoff–drainage–Δstorage terms (and layer-resolved soil moisture) to identify which term drives drift and how it changes across LAI experiments; otherwise the drift could arise from precipitation bias, runoff/drainage parameterization, soil texture/rooting depth, or stomatal stress representation.

We thank the reviewer for this important comment. We agree that the formulation referring to an "underestimation of LAI" as the cause of a multi-year spurious drying trend was misleading and incorrect. The drying trend is already present in the standard MAR configuration under present-day climate conditions, as illustrated in Fig. R1.

[Figure]

Figure R1 : Evolution of the soil moisture (expressed as a percentage of soil water saturation) between 2015 and 2022 in MAR$_{ref}$ and MAR$_{MODISclim}$.

What we intended to emphasize is that, if such a drying tendency already exists under current climate, then future simulations in which precipitation decreases in summer (Brajkovic et al., 2025), while vegetation is not allowed to decline accordingly, may artificially amplify this drying signal. In that case, the model would represent excessively dense vegetation relative to climate conditions, as illustrated by the MAR$_{+2\mu}$ experiment, which exhibits a stronger drying tendency than MAR$_{ref}$.

We therefore agree that the previous wording suggested an unsupported causal interpretation. To remove this ambiguity, Lines 506-509 will be rephrased to "The results suggest that the current MAR configuration, which relies on MERRA2 LAI, may underestimate the vegetation variability and density in Belgium. Although a drying trend is already present in the standard MAR configuration under current climate conditions while underestimating the vegetation cover, an inadequate representation of future vegetation changes may amplify this tendency. As in MAR$_{+2\mu}$, by maintaining unrealistically high vegetation density under potentially drier climate (Brajkovic et al., 2025), the drying trend could be exacerbated, thereby potentially affecting the reliability of long-term climate projections."

9. The added value of 8-day LAI during extremes is asserted as "subtle" and sometimes non-significant; strengthen this with event-based verification (phase/timing error, peak bias, tail metrics like TXx/TX95p, ET percentiles), and

ensure that smoothing choices in the MODIS climatology (32-day moving average) are not inadvertently damping drought signals you aim to test.

We thank the reviewer for this suggestion. We agree that event-based verification would ideally strengthen the assessment of the added value of the 8-day LAI product during extremes. However, the relatively short study period (10 years) and the limited number and duration of extreme events do not provide a sufficiently robust statistical basis for tail metrics such as TXx, TX95p, or evapotranspiration percentiles. For this reason, we focused instead on summers characterized by high and low SPEI3 values.

We also acknowledge that the 32-day moving-average smoothing applied in the construction of the MODIS climatology dampens drought-related variability. Consequently, the contrast with the unsmoothed 8-day product is expected to be even larger, reinforcing the interpretation.

To further strengthen this analysis, we will add two complementary panels to figure 11 focusing on shorter event periods and restricted to low-vegetation classes, which exhibit the strongest sensitivity. These additions will provide a more targeted illustration of the added value of the 8-day LAI product during extreme conditions.

[Figure]

Figure 11 : Taylor diagram for MAR$_{ref}$, MAR$_{MODIS}$, and MAR$_{MODISclim}$ for the 15 days with the higher/lower mean SPEI in July and August of drought (2018 (•), 2019 (+) and, 2022 (♦)), and pluvial years (2021 (▼), and 2023 (▲)) in low vegetation pixels. (c) Taylor diagram for average daily evapotranspiration (ET) (d) Taylor diagram for daily maximum surface air temperature. Points of the same colour are values from the same model while points of the same shape represent the same summer.

10. Reproducibility is insufficient for a sensitivity paper: key implementation choices (aggregation from 500 m to 5 km; sector masking rules; handling of missing

MODIS pixels; temporal interpolation step usage in SISVAT; any LAI caps) must be described so the experiments can be repeated and the sensitivity attributed to defined perturbations rather than undocumented preprocessing.

We thank the reviewer for this important comment regarding reproducibility. We agree that key preprocessing and implementation choices must be explicitly documented to ensure that the sensitivity experiments can be fully reproduced. The manuscript will therefore be revised to clarify all relevant steps, as detailed below.

**Spatial aggregation** : MODIS LAI values are resampled to the MAR 5 km grid by averaging all MODIS pixels partially or totally included within a MAR pixel, using weights that decrease linearly with the distance to the MAR pixel. Line 202 will be revised to:
  "MODIS LAI values are resampled to match the MAR 5 km grid by computing the weighted average of all MODIS pixels partially or totally included within each MAR pixel, with weights decreasing linearly with distance from the MAR pixel."

**Sector masking** :We are not entirely certain which part of the manuscript the reviewer refers to with "sector masking rules.. The term "sector" in MAR refers to subpixels representing different vegetation types within a single grid cell, analogous to the use of multiple plant functional types in a single pixel in other models.

**Handling of missing MODIS pixels** : We use the MODIS MCD15A2H Version 6.1 product, which selects the best-quality pixel from Terra and Aqua acquisitions within each 8-day period, strongly limiting missing values. The climatology contains no gaps due to temporal averaging over 10 years. During resampling of the 8-day product, no missing-pixel errors occurred; however, if such a case were to arise, the LAI value of the nearest available MODIS pixel would be used.

Line 198 will be revised to:
  "For this study, we replaced the MERRA2 LAI by the 8-day LAI composite from the MCD15A2H Version 6.1 MODIS Level-4 product (Myneni et al., 2021), which provides higher spatial (500 m) and temporal resolution. The product selects the best available pixel from Terra and Aqua acquisitions within each 8-day period."

**Temporal interpolation in SISVAT.** The temporal linear interpolation in SISVAT is a MAR time step interpolation between two 6 hourly time steps linearly derived from daily (MERRA) or weekly (MODIS) LAI values. Line 117 will be revised to:
  "The annual vegetation cycle is generated by linearly interpolating the input LAI values, which are provided daily from MERRA or weekly from MODISs (see Section 2.3)."

**LAI caps and clipping.** As discussed in our response to Comment #3, LAI caps and clipping associated with the Gaussian perturbations will be explicitly described in the methodology and discussed in the context of their impact on model sensitivity.

**Minor comments:**

1. Replace ambiguous phrasing ("cumulative mean daily") with explicit definitions (e.g., "seasonal total (mm)", "mean daily (mm/day)", "domain-mean daily"). Ensure every figure/table uses consistent units and labels.

We thank the reviewer for this comment. As stated in our response to Major Comment #1, all ambiguous phrasing such as "cumulative mean daily" will be replaced by explicit and consistent terminology, and all figures and tables will be checked to ensure consistent units and labels throughout the manuscript.

2. Clarify the soil moisture variable (volumetric water content vs saturation fraction), which soil layers are included (top layer vs root zone vs integrated 0–7 m), and whether values represent sector-weighted or grid-cell means.

Concerning soil moisture, the variable used in this study corresponds to the ratio between the soil water content and the soil water saturation integrated over the full 7 m soil column.

All vegetation-related variables in MAR are sector-weighted, meaning they are computed for each vegetation subpixel and then weighted by the fractional coverage of each vegetation type within the grid cell. The fractional coverage and vegetation type are fixed during the whole simulation.

Line 261 will be extended as follows:
 "For vegetation-related variables, if a specific vegetation type is indicated, the variable value refers only to the corresponding subpixel. If no vegetation type is specified, values represent grid-cell averages weighted by the fractional coverage of the different vegetation sectors. For soil-related variables, values are integrated over the top  7 m of soil."

3. Provide explicit sign conventions for surface fluxes (sensible heat in particular), since "positive toward the surface" differs from common conventions and affects interpretation.

In MAR, latent and sensible heat fluxes follow a convention in which positive values correspond to downward fluxes (toward the surface) and negative values to upward fluxes (toward the atmosphere), which is different from common conventions.

Line 325 will be extended as follows : "For sensible and latent heat flux, unlike other fluxes, a positive value indicates a flux directed toward the Earth's surface."

4. Specify whether "evaporation" includes canopy interception evaporation; if possible, report ET partition (transpiration, soil evaporation, interception) because LAI primarily changes transpiration and interception pathways.

As explained in Comment #1, Section 3.1 will be expanded to clearly define the physical variables used in this study. In particular, we will explicitly distinguish evaporation from evapotranspiration. Evaporation includes canopy-intercepted water, in which case an increase in LAI enhances both transpiration and evaporation (interception). However, the ET partition is not available directly as outputs from the model.

5. The SPEI description mixes "SPEI3" and "90-day"; use standard notation (e.g., SPEI-3) and explain the averaging/selection procedure for events more precisely (thresholds, windowing, persistence).

For the SPEI notation, we will adopt the standard terminology SPEI-3 throughout the manuscript and remove the ambiguous reference to "90-day." The event selection procedure will be clarified as follows. Drought periods are defined when the mean daily SPEI-3 averaged over Belgium during summer (JJA) is below −1.5, and pluvial periods when this value is above 1.

Accordingly, the Figure 2 caption will be revised to:
"Evolution of the SPEI-3 averaged over Belgium for the study period […]"

Line 170 will be revised to:
"The events were selected using the mean daily SPEI-3 index averaged over Belgium during summer (JJA). Drought events (mean daily summer SPEI-3 < −1.5) occurred in summers 2018, 2019, and 2022, while pluvial periods (mean daily summer SPEI-3 > 1) were observed in summers 2021, 2023, and 2024 (Fig. 2)."

6. The MODIS climatology period (2012–2022) overlaps drought years and differs from evaluation years; note the potential bias explicitly and, if feasible, add sensitivity to climatology period choice.

We acknowledge that the 2012–2022 period overlaps with drought years and differs from the evaluation period (2015–2024), which may introduce a potential bias. We therefore computed a climatology over 2015–2024 (Fig. R2) and found only negligible differences in LAI relative to the 2012–2022 climatology. This will be explicitly stated in the manuscript as follows (line 193):

"While this climatology period differs from the studied period (2015–2024) and may introduce potential bias, the resulting LAI values are very similar to those obtained using a climatology computed over 2015–2024 (see Supplementary Fig. R2)."

[Figure]

Figure R2: Comparison of MODIS LAI averages calculated using data from 2012–2022 versus 2015–2024.

7.  Quantify the impact of MODIS quality filtering and gap handling on spatial sampling (cloud-driven biases), especially in winter/spring.

As explained in Comment #10, we used the MODIS MCD15A2H Version 6.1 Level-3 product, which already applies quality filtering and selects the best available pixel from Terra and Aqua acquisitions within each 8-day composite, thereby strongly limiting missing values due to cloud contamination.

To quantify the impact of quality filtering and gap handling on the spatial sampling, we analysed the number of MODIS 500-m pixels contributing to each 5-km MAR grid cell. Figure R3 shows the distribution of the number of contributing MODIS pixels. More than 95% of MAR cells are computed from more than 80 MODIS pixels, indicating a very robust spatial sampling even in winter and spring.

The few MAR grid cells with a lower number of contributing MODIS pixels correspond mainly to pixels classified as urban by MODIS but not by MAR, for which no LAI value is provided in the MODIS product. These pixels therefore do not reflect cloud-driven data gaps.

Consequently, we conclude that cloud-driven quality filtering and gap handling have a negligible impact on the spatial representativeness of the MODIS LAI input in our study domain.

[Figure]

Figure R3 : Distribution of the number of contributing MODIS pixels to each MAR pixel for the 2015-2024 period.

8. Figure interpretations would benefit from stratification by land-cover class (forest/crops/grass) beyond national averages; many responses (albedo, ET) are class-dependent.

We agree that stratifying the analysis by land-cover class would provide additional insight into the class-dependent responses. This approach is already applied in Section 3.5, where vegetation type differentiation where the LAI forcing is based on observational data rather than the biased Gaussian perturbations. In addition, as noted in our response to Comment #9, we will add complementary figures to Figure 11 focusing on shorter event periods and restricted to low-vegetation classes, which exhibit the strongest sensitivity. These additions will provide a more targeted illustration of the added value of the 8-day LAI product during extreme conditions as a function of vegetation type.

9. Where you mention "model instability" (MAR−2μ), provide diagnostics (runoff, drainage, soil moisture bounds, energy closure) rather than a qualitative statement.

We clarify that the reported instability in the MAR−2μ simulation is of numerical rather than physical origin. When LAI approaches zero, some model formulations involve divisions by very small values, leading to unrealistically large values and causing the simulation to diverge. To avoid ambiguity, the manuscript will be revised accordingly.

Line 233 will be changed to:

 "The MAR$_{-2\mu}$ simulation was stopped in 2019 as the model began to drift away from observations due to numerical instability emerging in 2020 and affecting the hydrological budget."

Line 363 will be changed to:

 "The largest changes were observed in the MAR$_{-2\mu}$ simulation, which may explain the subsequent numerical instability of the model."

10. Clean up typos/formatting (missing spaces, "n response", duplicated punctuation) and ensure references are consistent (e.g., IPCC citation formatting).

The authors would like to thank the reviewer for noticing the typos. The text will be reviewed and the typos and references will be changed accordingly.

**Bibliography**

Brajkovic, J., Fettweis, X., Noël, B., Vyver, H. V. D., Ghilain, N., Archambeau, P., Pirotton, M., and Doutreloup, S. (2025). Increased intensity and frequency of extreme precipitation events in Belgium as simulated by the regional climate model MAR, Journal of Hydrology: Regional Studies, 59, 102 399, https://doi.org/10.1016/j.ejrh.2025.102399.

De Ridder, K. (1997). Radiative Transfer in the IAGL Land Surface Model. Journal of Applied Meteorology, 36(1), 12-21. https://doi.org/10.1175/1520-0450(1997)036<0012:RTITIL>2.0.CO;2

Doutreloup S. (2022). Évolution actuelle et future des précipitations convectives sur la Belgique et la région du Lac Victoria (Afrique équatoriale de l'Est) à l'aide du modèle climatique régional MAR. Climatologie, 19, 1.